# FEM Simulation of THz Detector Based on Sb and Bi$_{88}$Sb$_{12}$ Thermoelectric Thin Films

**Anastasiia S. Tukmakova** [1,*] , **Alexei V. Asach** [1] , **Anna V. Novotelnova** [1] ,
**Ivan L. Tkhorzhevskiy** [1] , **Natallya S. Kablukova** [2,3] , **Petr S. Demchenko** [4] ,
**Anton D. Zaitsev** [4] **and Mikhail K. Khodzitsky** [4]

1 Faculty of Cryogenic Engineering, ITMO University, 197101 Saint-Petersburg, Russia;
avasach@itmo.ru (A.V.A.); novotelnova@itmo.ru (A.V.N.); tkhorzhevskiy.ivan.l@gmail.com (I.L.T.)
2 International Scientific and Research Institute of Bioengineering, ITMO University, 197101 Saint-Petersburg,
Russia; kablukova.natali@yandex.ru
3 The Department of Physics, Saint Petersburg State University of Industrial Technologies and Design,
191186 Saint-Petersburg, Russia
4 Terahertz Biomedicine Laboratory, ITMO University, 197101 Saint-Petersburg, Russia;
petr.s.demchenko@gmail.com (P.S.D.); a.zaytsev@niuitmo.ru (A.D.Z.); khodzitskiy@yandex.ru (M.K.K.)
* Correspondence: astukmakova@itmo.ru

**Abstract:** A terahertz (THz) detector based on thermoelectric thin films was simulated using the finite elements method. The thermoelectric circuit consisted of *Sb* and *Bi$_{88}$Sb$_{12}$* 150-nm films on the mica substrate. *Sb*, *Bi$_{88}$Sb$_{12}$*, and mica-substrate properties have been measured experimentally in the THz frequency range. The model of electromagnetic heating was used in order to estimate possible heating of *Sb-Bi$_{88}$Sb$_{12}$* contact. THz radiation power varied from 1 μW to 50 mW, and frequency varied in the range from 0.3 to 0.5 THz. The calculations showed a temperature difference of up to 1 K, voltage up to 0.1 mV, and responsivity of several mVW$^{-1}$. The results show that thin *Sb* and *Bi − Sb* thermoelectric films can be used for THz radiation detection at room temperatures.

**Keywords:** thermoelectric; detector; terahertz; FEM; finite element simulation; bismuth; antimony

## 1. Introduction

Terahertz (THz) devices are of significant importance for such fields as disease diagnostics and medical imaging [1–3], security systems [4,5], and sensing and imaging [6]. Thermal THz sensors, such as bolometers, are used for THz detection [7]. The effect of electromagnetic heating due to THz radiation absorption can be used as the signal for thermoelectric (TE) detector. TE detectors work on the basis of the Seebeck effect, which generates electric voltage in TE circuits in case of temperature difference between two junctions (hot and cold junctions) of TE materials [8,9]. Basically, the temperature of cold junction is maintained at room temperature, while hot junction temperature is to be increased due to the electromagnetic heating. In literature, the effect combining optical and thermoelectric phenomena is called photothermoelectric, and it has been studied with the use of different advanced materials: graphene [10], single-layered *MoS$_2$* [11], *Bi$_2$Se$_3$* nanoribbons [12], and organic thermoelectrics [13]. A line of works about THz detectors working on the TE effect have been published. In [14], a responsivity of several hundred VW$^{-1}$ was reported for a detector based on single-layer graphene on 300-nm *SiO$_2$* substrate. A THz TE detector based on carbon nanotubes with responsivity of several VW$^{-1}$ was reported in [15].

The increase of efficiency or responsivity of TE detector is possible due to the increase of temperature difference between junctions or due to the increase of TE figure of merit *Z*. *Z* is proportional

to Seebeck coefficient squared $S^2$ and electrical conductivity $\sigma$; $Z$ is inversely proportional to thermal conductivity $\kappa$. In order to increase temperature difference, the cross-section of hot junction can be decreased. The hot junction can be thermally insulated, for example, in vacuum, and the device can be encapsulated in order to prevent convection.

Besides of temperature difference increase, the usage of low-dimensional TE materials interconnects with Z-value improvement. It was shown that 2D and 1D thermoelectrics could give a significant increase of TE figure of merit [16]. In particular, TE thin films are known for the possibility to produce a high Seebeck coefficient and power factor $S^2 \cdot \sigma$. As all thermoelectric properties are the functions of temperature, not every material will be suitable for detecting purposes requiring near-room temperatures. TE films based on *Bi* and *Te* alloys are good candidates for the applications at room temperature, their properties have been reported in [17]. Other candidates are TE films based on *Bi* and *Sb* solid solutions. In a $Bi_{(1-x)}Sb_{(x)}$ system with Sb concentration lower than 15%, the Seebeck coefficient value at room temperatures is around −100 μV/K. This system was studied in the works of Volklein et al. in the 1990s [18] and Suslov et al. in 2017 [19]. Close values of Seebeck coefficient have been reported for the $Bi_{0.91}Sb_{0.09}$ system in [20]. In an article published by Linseis et al. in 2018 [21], a Seebeck coefficient of $Bi_{87}Sb_{13}$ was around 80–110 μV·K$^{-1}$ with comparatively low thermal conductivity (1.5–3.5 W·m$^{-1}$K$^{-1}$) and high electrical conductivity (around 100–300 kS·m$^{-1}$) at room temperatures. Sb thin film was used in our study as a possible p-type material, which can have a good agreement in electrical conductivity with Bi-Sb solid solutions. Seebeck coefficient of Sb is around 30 μV/K [22]. Moreover, the presence of a narrow band gap makes it possible for Bi-Sb to absorb THz radiation. The band structure of Bi-Sb may be seen in [23], the results have been taken from the work published in 1970 [24]. Surface band structure of $Bi_{(x-1)}Sb_{(x)}$ was reported in [25]. Hence, from the point of view of the highest Seebeck coefficient at room temperature and band structure, Bi-Sb thin films potentially can be used as a sensitive element for THz detection.

The bismuth-antimony thermocouple has already been considered as a possible basis for far-infrared detection in 1982 and showed relatively low noise equivalent power. In [26], an antenna based on *n*-type $Bi_{0.87}Sb_{0.13}$ and *p*-type Sb thin films on silicon substrate was proposed for 0.812-THz frequency detection. The simulations showed a possible temperature difference of up to 7.3 K in vacuum sealing under the input power of 7.5 μW. The responsivity was 30 *V/W* in the air and 119 *V/W* in vacuum for a pixel with 8 thermocouples. In [27], an antenna made of 8 $Sb - Bi_{87}Sb_{13}$ thermocouples was fabricated with 30-μm length and 2-μm width. It showed an output voltage of 0.1–0.7 μV. The main advantage of this antenna was the possibility of room-temperature operation, low noise equivalent power (from 100 to 1000 pW·Hz$^{-0.5}$), and a fast response of 22 μs.

Finite elements simulation method showed itself as a reasonable and effective tool for THz antenna modelling, making it possible to combine thermal, electrical, and optical phenomena in one multiphysical model. For example, in [28], THz-resonant antenna was simulated to investigate the photocurrent value as the function on the antenna length. The simulation results were in good agreement with the experimental measurements. In [26], an electromagnetic heating model was used in order to evaluate temperature increase due to THz radiation absorption. A line of works about FEM simulation of thermoelectric detectors of infrared radiation, which is close to THz range, has been presented in the literature [29].

The aim of this work is to simulate a THz detector based on Sb and $Bi_{88}Sb_{12}$ TE thin films in order to evaluate possible heating of hot junction due to the electromagnetic heating at 0.3, 0.4, and 0.5 THz. Such a detector can be potentially used as a pixel of THz imaging system.

## 2. Materials and Methods

### 2.1. Model Geometry and Mesh

3D model geometry is presented in Figure 1. Cross section in *x-z* plane is presented in Figure 2 and gives a detailed representation of antenna geometry. Position numbers (1–6) in both figures correspond

to the same elements. Antenna had a square *x-y* cross-section $1 \times 1$ mm$^2$. We used free tetrahedral user-controlled mesh, as the correlation between maximum mesh element size and wavelength must be controlled. Maximum element size was set equal to 40 µm.

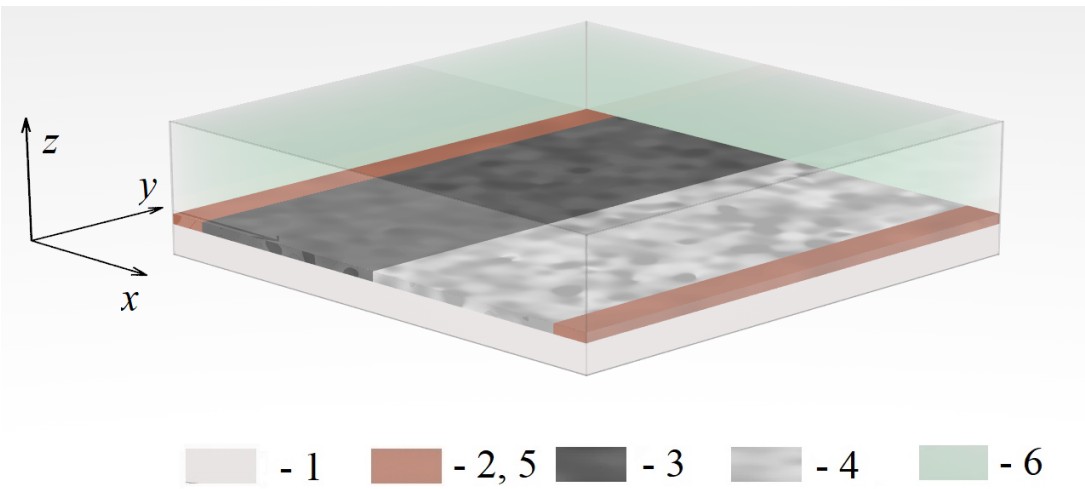

**Figure 1.** 3D view of simulated antenna. 1—mica substrate; 2,5—copper films, 3—$Bi_{88}Sb_{12}$ film, 4—$Sb$ film, 6—vacuum chamber.

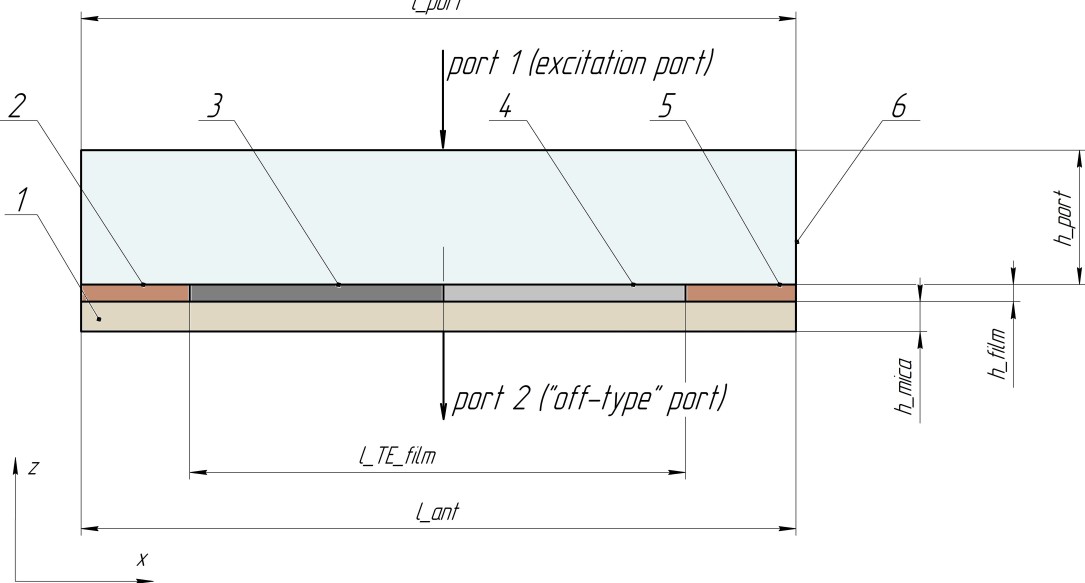

**Figure 2.** Antenna cross-section in *z-x* plane; $l_{port}$ = 1000 µm—the length of the wave excitation port; $l_{TEfilm}$ = 800 µm–the length of thermoelectric couple containing two films, $l_{ant} = l_{port}$—the full antenna length; $h_{film}$ = 150 nm—the thickness of the films; $h_{mica}$ = 40 µm—the thickness of mica substrate; $h_{port}$—the distance between wave excitation and antenna, the value of $h_{port}$ was changing within the simulations.

## 2.2. Mathematical Model

Interaction between detector and THz radiation was described by 2 interfaces: (1) electromagnetic heating and (2) heat transfer in solids. Radiation was excited from the port, passed through the vacuum, absorbed by TE films, and caused heating. The resulting temperature is the dependent variable to be found.

### 2.2.1. General Equations

**Wave equation:**

$$\nabla \times \mu_r^{-1}(\nabla \times \mathrm{E}) - k_0^2(\epsilon_r - i\sigma/(\omega\epsilon_0))\mathrm{E} = 0, \tag{1}$$

where $\mu_r$ is the relative permeability; E is the electric field; $k_0$ is the wave number in free space; $\epsilon_r$ and $\epsilon_0$ are the relative and free space permittivity, respectively; $\sigma$ is the electrical conductivity; and $\omega$ is the angular frequency.

For the electric field displacement description, the dielectric loss model was used:

$$\epsilon_r = \epsilon' - i\epsilon'', \tag{2}$$

where $\epsilon'$ and $\epsilon''$ are real and imaginary parts of permittivity.

**Heat transfer:**

$$\nabla q = -\kappa \nabla T = Q_e, \tag{3}$$

where q is the heat flux and $Q_e$ is the electromagnetic heat source, $T$ is the temperature, $\kappa$—thermal conductivity.

Electromagnetic heat source or electromagnetic loss density includes resistive and magnetic losses:

$$Q_e = Q_{rh} + Q_{ml} = \frac{1}{2}Re(\mathrm{J} \cdot \mathrm{E}^*) + \frac{1}{2}Re(i\omega\mathrm{B} \cdot \mathrm{H}^*), \tag{4}$$

where J is the electric current density, E is the electric field strength, B is the magnetic induction, and H is the magnetic field strength.

### 2.2.2. Boundary Conditions

The excitation port 1 (excitation port on Figure 2) was applied to the upper surface of the simulated system ($1 \times 1$ mm$^2$) and was described by the following equation:

$$S = \frac{\int_{\delta\Omega}(\mathrm{E} - \mathrm{E}_1) \cdot \mathrm{E}_1}{\int_{\delta\Omega}\mathrm{E}_1 \cdot \mathrm{E}_1}, \tag{5}$$

where $\mathrm{E}_1$ and $\mathrm{E}_2$ are electric fields on port 1 and port 2, respectively.

Port 2 ("off-type" port on Figure 2) was applied to the whole lower surface of the detector ($1 \times 1$ mm$^2$) corresponding to the mica-substrate downside. The "off-type" port was described by the following equation:

$$S = \frac{\int_{\delta\Omega}\mathrm{E} \cdot \mathrm{E}_2}{\int_{\delta\Omega}\mathrm{E}_2 \cdot \mathrm{E}_2}. \tag{6}$$

The scattering boundary condition was applied to the lateral surfaces:

$$\mathrm{n} \times (\nabla \times (\mathrm{E})) - ik\mathrm{n} \times (\mathrm{E} \times \mathrm{n}) = 0, \tag{7}$$

where n-normal vector.

### 2.2.3. Mathematical Description of Thermal and Electromagnetic Phenomena in Thin Films

Four thin-layer objects were in the model: TE thin films of *Sb* and *Bi*$_{88}$*Sb*$_{12}$, and two copper films. The thickness of all films was 150 nm. In order to reduce the model size and computational time, analytical expressions were used to described thin objects.

In electromagnetic interfaces, a dielectric loss model is required. For this purpose, a transition boundary condition was used:

$$\mathrm{J}_{s.up} = \frac{Z_s\mathrm{E}_{t.up} - Z_t\mathrm{E}_{t.down}}{Z_s^2 - Z_t^2}, \tag{8}$$

$$J_{s.down} = \frac{Z_s E_{t.down} - Z_t E_{t.up}}{Z_s^2 - Z_t^2}, \tag{9}$$

$$Z_s = \frac{-i\omega\mu}{k} \frac{1}{tan(kd)}, \tag{10}$$

$$Z_t = \frac{-i\omega\mu}{k} \frac{1}{sin(kd)}, \tag{11}$$

$$k = \omega\sqrt{\epsilon_r + (\sigma/(i\omega))\mu_r}, \tag{12}$$

where indices "*up*" and "*down*" correspond to upsides and downsides of the layer, $Z_s$—surface impedance, $Z_t$—tangential impedance, $J_s$—surface current density, and $E_t$—tangential electric field.

In the heat transfer interface, a thin layer description was used in order to describe heat transfer in thin objects:

$$\nabla_t q_s = d_s Q_s + q_0, \tag{13}$$

$$q_s = -d_s \kappa \nabla_t T, \tag{14}$$

where $d_s$ is film thickness, $Q_s$ is a density-distributed heat source, $q_0$—heat flux received out-of-plane, $q_s$—heat flux in the film, and $\nabla_t$ stands for tangential derivative.

Thermally thin approximation model was used for the layer description. This model is valid for films with high thermal conductivity, or conductivity which is higher than the surroundings.

## 2.3. Materials

The films were prepared by the thermal vacuum deposition with the subsequent annealing. This method was used for $Bi_{x-1}Sb_x$-films preparation and was described in [30]. The properties of *Sb* and $Bi_{88}Sb_{12}$ films of 150-nm thickness have been chosen for simulation. Electrical conductivity and relative permittivity of mica substrate, *Sb*, and $Bi_{88}Sb_{12}$ thin films in terahertz frequency range were measured using a terahertz time-domain spectrometer (THz-TDS) in thin-film approximation [31] (when a wavelength of THz radiation is by orders of magnitude higher than the film thickness). The waveforms of THz single-pulse (temporal dependencies of THz signal electric field amplitude) were measured for an air (reference signal), mica substrate, and a film. The corresponding amplitude and phase spectra were obtained in the 0.3–0.8 THz frequency range through the Fourier transform method. The complex sheet conductivity of a film was calculated as

$$\hat{\sigma}(f) = [(\hat{n}_{sub}(f) + 1)\hat{E}_{sub}(f)/\hat{E}_{sam}(f) - \hat{n}_{sub}(f) - 1]/Z_0, \tag{15}$$

where $f$ is the frequency of THz radiation, $Z_0$ is the free-space impedance, $\hat{E}_{sub}$ and $\hat{E}_{sam}$ are complex amplitudes of THz wave transmitted through the substrate and the sample, and $\hat{n}_{sub}$ is the substrate complex refractive index—which is calculated as

$$\hat{n}_{sub}(f) = c[\phi_{sub}(f) - \phi_{air}(f)]/(2\pi f d_{sub}) + 1 - i\, c\, ln(|\hat{E}_{sub}^2(f)|/|\hat{E}_{air}^2(f)|)/(4\pi f d_{sub}), \tag{16}$$

where $c$ is the speed of light, $\phi_{sub}$ and $\phi_{air}$ are phases of substrate and air signals, $d_{sub}$ is the substrate thickness, and $\hat{E}_{air}$ is the complex amplitude of the THz wave transmitted through the air. The complex film permittivity was extracted from the complex conductivity as

$$\hat{\varepsilon}(f) = 1 + i\hat{\sigma}(f)/(2\pi f d_{film}\varepsilon_0), \tag{17}$$

where $d_{film}$ is the film thickness and $\varepsilon_0$ is a permittivity of free space.

The results of measurements are shown in Figures 3–5. It should be mentioned that they are in good enough agreement with the classical Drude model (for conductivity $\hat{\sigma}$ and permittivity $\hat{\varepsilon}$) due to the semimetallic nature of the films:

$$\hat{\sigma}(\omega) = \frac{\sigma_0}{1 - i\omega\tau},$$ (18)

$$\hat{\varepsilon}(\omega) = \varepsilon_\infty - \frac{\omega_p^2}{\omega^2 + i\gamma\omega},$$ (19)

where $\sigma_0$ is the DC conductivity, $\omega = 2\pi f$ is the angular frequency of THz radiation, $\tau$ is the relaxation time of charge carriers, $\varepsilon_\infty$ is high-frequency dielectric constant, $\omega_p$ is the plasma frequency, and $\gamma$ is the damping rate.

Material properties used in the heat transfer interface are listed in Table 1.

**Table 1.** Thermophysical properties of the materials.

| Material | Thermal Conductivity, in-Plane, $W \cdot m^{-1} \cdot K^{-1}$ | Thermal Conductivity, Cross-Plane, $W \cdot m^{-1} \cdot K^{-1}$ | Heat Capacity, $J \cdot kg^{-1} \cdot K^{-1}$ | Density, $kg \cdot m^{-3}$ |
|---|---|---|---|---|
| Sb | 63.24 | 24.43 | 207 | 6691 |
| $Bi_{88}Sb_{12}$ | 13.21 | 5.1 | 124 | 9790 |
| mica | 5.1 | 0.51 | 880 | 2900 |
| Cu | 401 | 401 | 384 | 8960 |

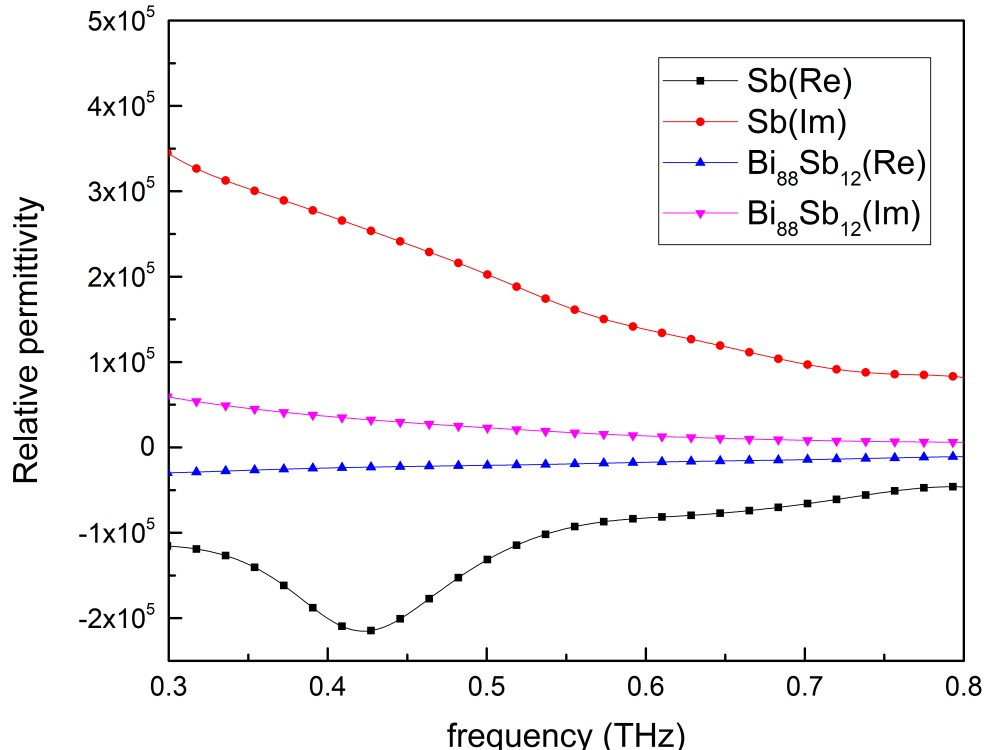

**Figure 3.** The real and imaginary parts of a relative permittivity of thin-film Sb and $Bi_{88}Sb_{12}$ on mica substrate in the frequency range of 0.3–0.8 THz.

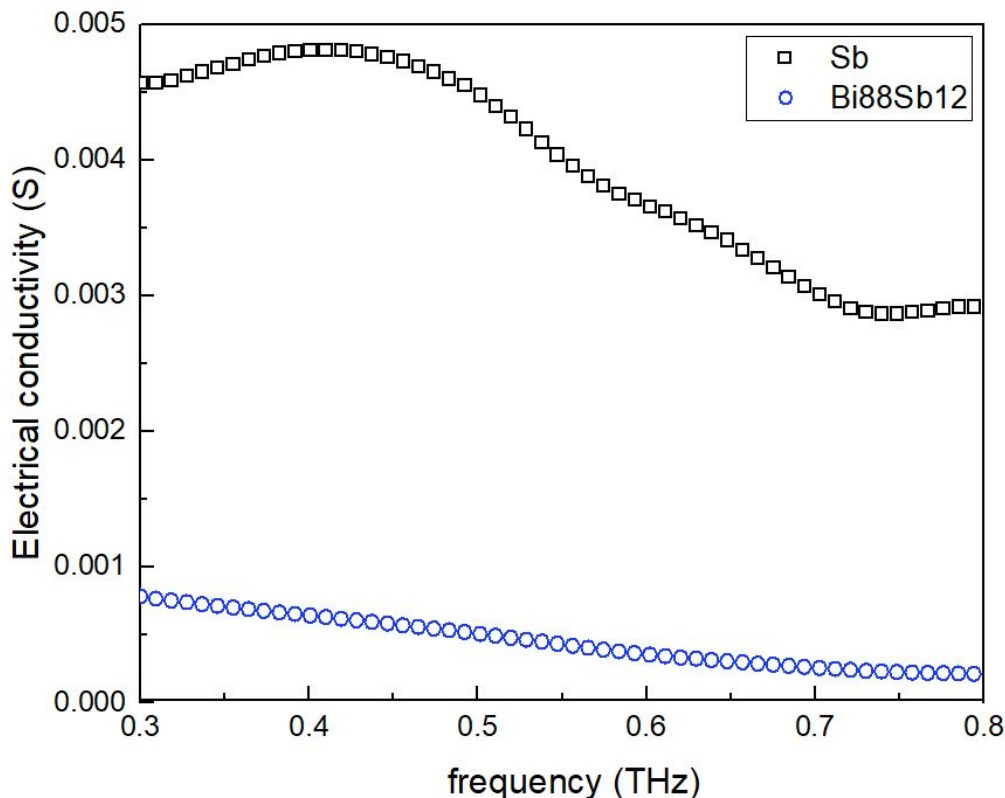

**Figure 4.** Electrical conductivity of antimony and bismuth antimonide 150-nm thin films on mica substrate in the frequency range of 0.3–0.8 THz.

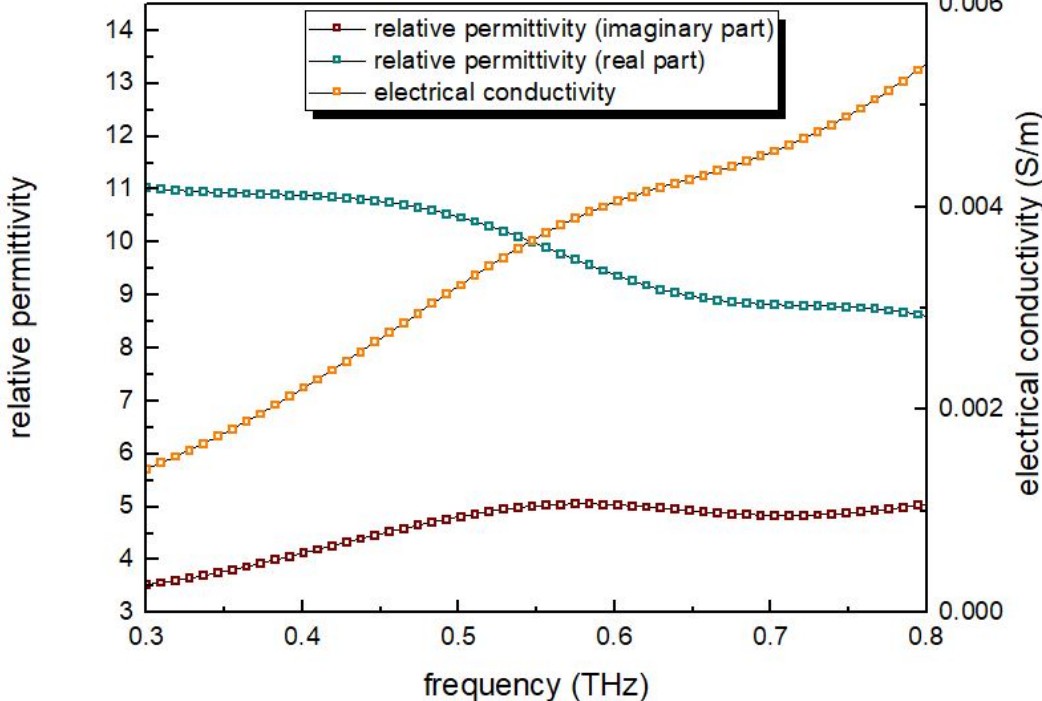

**Figure 5.** The properties of mica substrate used in the simulation in the frequency range of 0.3–0.8 THz.

The real part of relative complex permittivity is strongly negative for both the films in the studied frequency range, in accordance with the Drude model, for metals/semimetals below the plasma frequency. The behaviour of electrical conductivity also follows this model; generally, it decreases

with increasing frequency of THz radiation. The electrical conductivity is higher for pure stibium film, as well as absolute values of real and imaginary parts of its permittivity. The mica has dielectric properties, so it's permittivity does not follow the Drude theory.

In order to evaluate the dispersion of copper-relative permittivity, the Drude–Lorentz model for metals was used [32]. Calculated real (Re) and imaginary (Im) parts of relative permittivity are shown in Table 2.

**Table 2.** Calculated relative permittivity of copper.

| Frequency, THz | Re | Im |
|:---:|:---:|:---:|
| 0.1 | −6219.67 | $2.70 \times 10^2$ |
| 0.2 | −6219.42 | $1.35 \times 10^2$ |
| 0.3 | −6218.99 | $9.00 \times 10$ |
| 0.4 | −6218.39 | $6.75 \times 10$ |
| 0.5 | −6217.62 | $5.40 \times 10$ |
| 0.6 | −6216.68 | $4.50 \times 10$ |
| 0.7 | −6215.58 | $3.86 \times 10$ |
| 0.8 | −6214.3 | $3.37 \times 10$ |

## 3. Results

### 3.1. Temperature Distribution Along the Antenna Length

The distribution of temperature difference between absolute temperature and ambient temperature $T_{amb}$ = 293.15 K is presented in Figure 6. Hot junction corresponds to $x$-coordinate at 500 μm. The $Bi_{88}Sb_{12}$ layer is heated up to a higher temperature in comparison with the $Sb$ layer. Temperature in the $Bi_{88}Sb_{12}$ layer is higher than in the hot junction. The impact of heating in the $Cu$ layer is almost negligible. The results are presented for different $h_{port}$ values. The same picture of temperature distribution along the antenna length was true for all simulation cases (with different power input and frequencies).

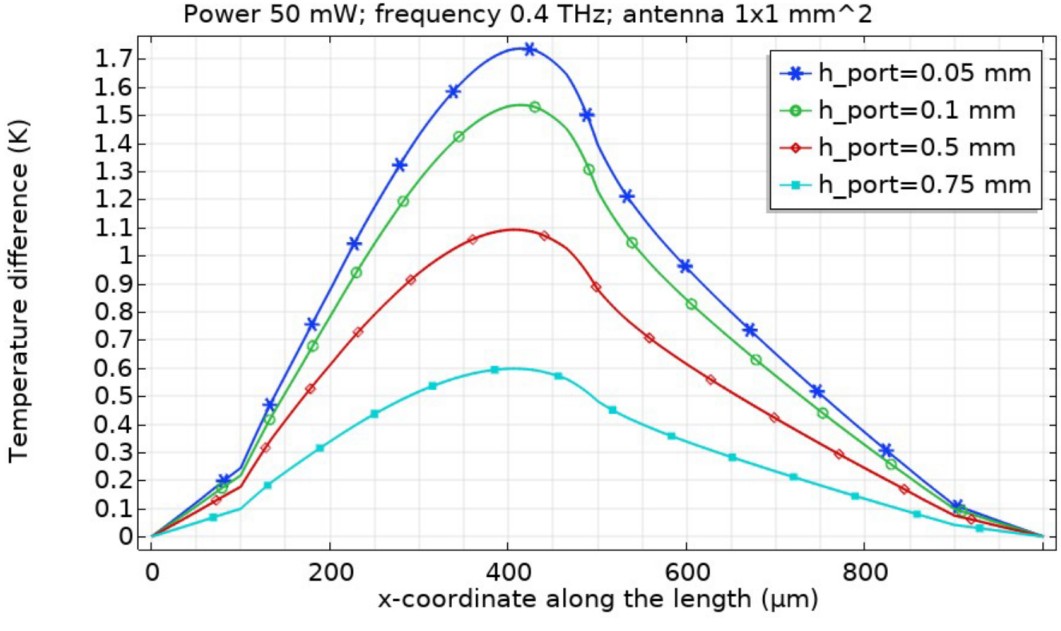

**Figure 6.** Example of temperature distribution along the x-axis (antenna length). Copper films correspond to the x-coordinates: 0–100 μm, $Bi_{88}Sb_{12}$ to 100–500 μm, $Sb$ to 500–800 μm; hot junction corresponds to the x-coordinate at 500 μm.

### 3.2. Temperature Difference and Responsivity

One of the main goals of this paper is to evaluate the responsivity of the detector. For this purpose, one should know the amount of power scattered and transferred to the detector working surface. The amount of power absorbed will differ from the power on the excitation port. The difference between excitation power and power on the detector surface can be expressed in terms of S-parameter (scattering parameter). The part of S-parameter expressing the transmission is $S_{21}$. The $S_{21}$ value differs within the change of $h_{port}$ and has a dependence on frequency. Parameter $S_{21}$ was evaluated using the following expression:

$$S_{21} = \frac{\int_{port2}(E_c \cdot E_2^*)dA_2}{\int_{port2}(E_2 \cdot E_2^*)dA_2},$$ (20)

where $E_c$—the computed electric field which consists of the excitation and reflected field, $A_2$ is the port 2 surface area, and * symbol means that the calculation is to be made at a specific mode number.

The amount of power $P$ transmitted to the material can be found in terms of the following expression:

$$P = P_{ex} \cdot S_{21}^2,$$ (21)

where $S_{21}$—transmission parameter, $P_{ex}$—emitted power.

In Figure 7, the square of $S_{21}$ absolute value is presented. The results for different $P_{ex}$ values were equal. In our case, distances between wave excitation port and antenna from several tens of microns to several millimetres are allowable. At the distance equal to 2 mm, the part of scattered radiation is from 0.81 to 0.97. The wave with the frequency of 0.3 THz scatters more intensively than with 0.4 and 0.5 THz. The lowest scattering rate can be seen at the frequency of 0.5 THz.

The oscillations in transmission parameter dependence on the distance between THz radiation source and detector are related to wave interference between source and detector. The troughs in the transmission occur at distances given by $h_{port} = (c/(2f)) \cdot N$, where $c$ is the speed of light, $f$ is the frequency, and the integer $N$ is the order of the trough.

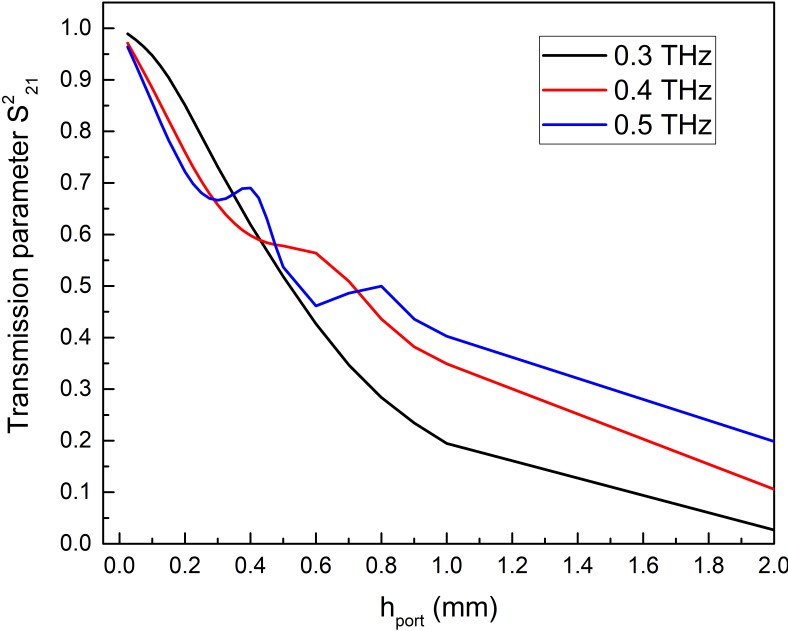

**Figure 7.** Transmission parameter dependence on the distance between THz radiation source and detector. The results are obtained for the power input of 50 mW.

The transmission parameter decreases with the increase of the distance between wave excitation port and antenna. Hence, the resulting hot junction temperature $T_h$ must be inversely proportional to the $h_{port}$. In Figures 8–10, the dependence of temperature difference $\Delta T_h$ between hot junction and ambient temperature on $h_{port}$ is presented. Hot junction temperature was calculated as the average value within the hot contact length.

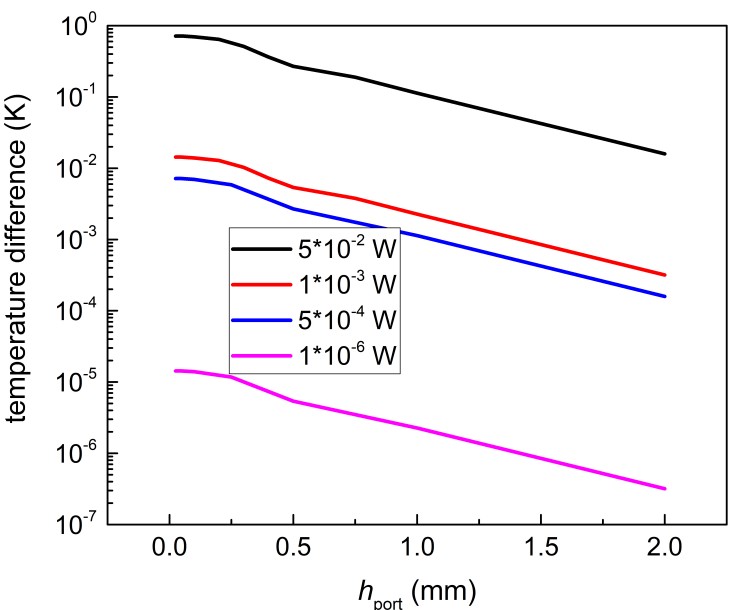

**Figure 8.** Temperature difference on $h_{port}$ at 0.3 THz; $P_{ex}$ values are listed in the legend.

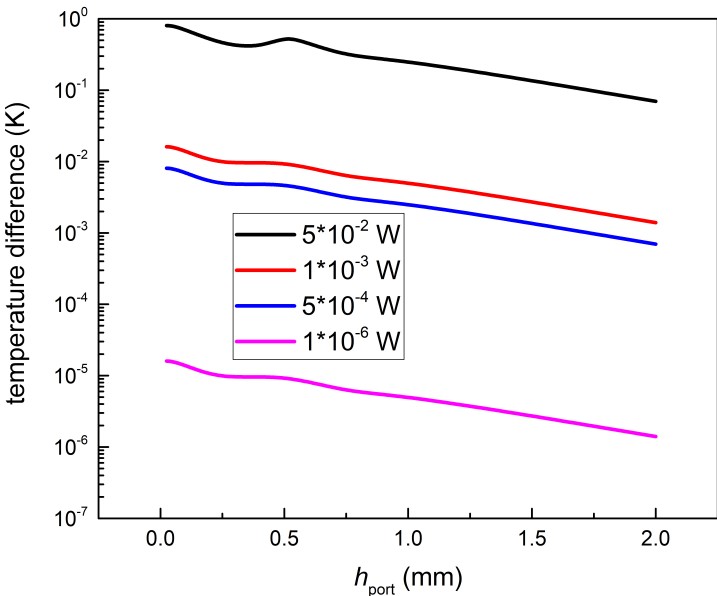

**Figure 9.** Temperature difference on $h_{port}$ at 0.4 THz; $P_{ex}$ values are listed in the legend.

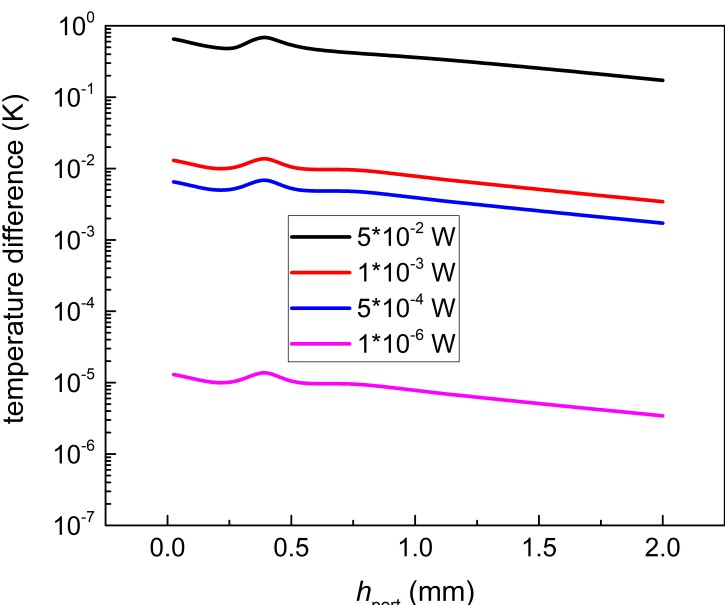

**Figure 10.** Temperature difference on $h_{port}$ at 0.5 THz; $P_{ex}$ values are listed in the legend.

The responsivity of the antenna at different frequencies is presented in Figure 11. The dependence is nonlinear for all the frequencies and changes its value from 1 to 2.5 mVW$^{-1}$. The responsivity at 0.3 THz has the lowest values, no higher than 1.5 mVW$^{-1}$. The responsivity at 0.5 THz is the highest—from 1.25 to 2.5 mVW$^{-1}$. No dependence of responsivity on the excitation power was observed.

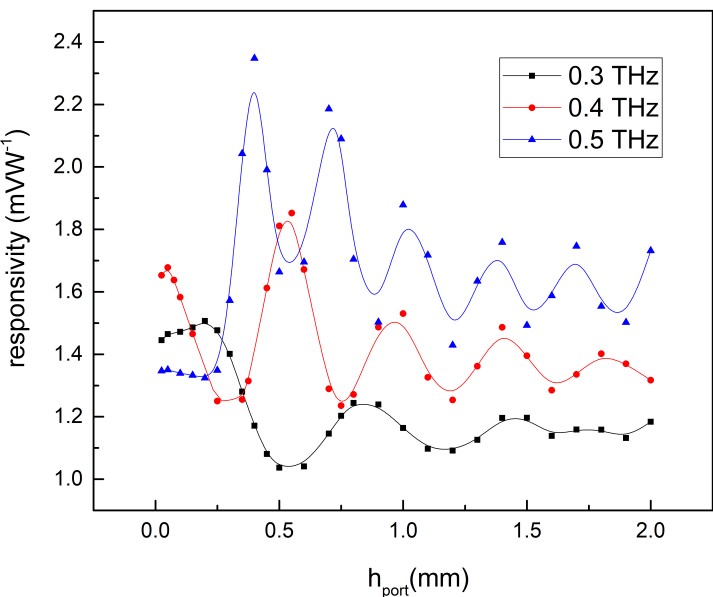

**Figure 11.** Responsivity of antenna on $h_{port}$ in frequency range from 0.3–0.5 THz.

## 4. Discussion

Temperature difference between hot and cold junctions in TE circuit was investigated. Voltage in TE circuit is found as $\Delta V = (|S_n| + |S_p|) \cdot \Delta T$. $Sb$-$Bi_{88}Sb_{12}$ circuit can produce $(|S_n| + |S_p|)$ of around 100 µV/K. For an excitation port power of 500 µW, a possible response is expected to be around 1 µV. At lower power (e.g., $P_{ex}$ =1 µW) the expected response is lower. At $P_{ex}$ from 1 to 50 mW, the voltage is expected to be from 10 µ V to 0.1 mV, which is possible to detect.

The calculated responsivity is from 1 to 2.5 mVW$^{-1}$. The increase in frequency leads to the increase in responsivity. Higher values of TE Seebeck coefficients of thermocouple (up to 300 $VK^{-1}$) can increase the responsivity by 2–3 times. The dependence of responsivity on the distance between excitation port and detector has smoother behaviour when the $h_{port}$-value is higher than 1 mm.

Thin layer of $Bi_{88}Sb_{12}$ showed higher sensitivity to THz radiation with higher resulting temperatures. The temperature of hot junction is lower than in the $Bi_{88}Sb_{12}$ layer. Hence, it is possible to do a thermoelectric circuit containing one TE material—$Bi_{88}Sb_{12}$ and metal, for example $Ni$. Due to the rather high temperature increase (1-2 K at single $Bi_{88}Sb_{12}$) and the significant Seebeck coefficient (around 100 µVK$^{-1}$), a voltage of around 0.1 mV can be expected in such a circuit.

Moreover, other charge-density-wave compounds with high thermoelectric properties at room temperature may be potentially used for THz-detecting purposes. Thin films of well-known thermoelectrics based on Bi, Te, Sb, Sn, and their compounds are of great interest for such applications.

**Author Contributions:** Conceptualization, M.K.K., N.S.K., and A.S.T.; methodology, A.V.A., A.S.T., N.S.K., and P.S.D.; formal analysis, A.S.T., A.V.A., and A.V.N.; investigation, A.S.T., P.S.D.; data curation, A.D.Z. and A.V.N.; writing—original draft preparation, A.S.T. and A.V.A.; writing—review and editing, M.K.K.; visualization, I.L.T.; supervision, M.K.K.; funding acquisition, M.K.K. All authors have read and agreed to the published version of the manuscript.

**Funding:** This research was funded by Russian Science Foundation, grant number 19-72-10141.

**Conflicts of Interest:** The authors declare no conflict of interest. The funders had no role in the design of the study; in the collection, analyses, or interpretation of data; in the writing of the manuscript, or in the decision to publish the results.

## Abbreviations

The following abbreviations are used in this manuscript:

TE　　Thermoelectric
THz　 Terahertz
Sb　　Antimony
Bi　　Bismuth
Ni　　Nickel

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
