# Peer review of "FEM Simulation of THz Detector Based on Sb and Bi88Sb12 Thermoelectric Thin Films"

_applsci, doi:10.3390/app10061929_

Round 1

Reviewer 1 Report

The authors provide simulation using the FEM method for THz detectors utilizing a TE thin film. More other recent works in the similar technique can be introduced in addition to [20], which was published in 2012. 

The locations of port 1 and 2 in FEM simulation is unclear. 

In Table 2, why are the real parts of the relative permittivity for copper negative?

In Fig. 7, the transmission of 0.5 THz data appears to be non-monotonic, different from other data. Can the authors give more discussions on this?

Author Response

   1. The authors provide simulation using the FEM method for THz detectors utilizing a TE thin film. More other recent works in the similar technique can be introduced in addition to [20], which was published in 2012. 

Re: We have added more analysis and one more reference about Bi-Sb THz antenna (please, see lines 48-55). Also we added an additional paragraph (lines 56-62) with the analysis of FEM simulation of THz and microwave antennas (links [24, 26-27] to prove that this method is actual and effective.  

    2. The locations of port 1 and 2 in FEM simulation is unclear. 

Re: figure 2 has been completed with the ports location designation. We also added the clarifications to the text to the section 2.2.2 (lines 85-87).

    3. In Table 2, why are the real parts of the relative permittivity for copper negative?

Re: It is negative in accordance with Drude formalism. Real part of relative permittivity is negative and slightly increases with the increase of the frequency. Imaginary part is positive and decreases with frquency increase. 

    4. In Fig. 7, the transmission of 0.5 THz data appears to be non-monotonic, different from other data. Can the authors give more discussions on this?

Re: This effect can be seen on all the ferquencies. The oscillations in transmission dependence on the distance between THz radiation source and detector are related to wave interference between source and detector. The troughs in the transmission occur at distances given by L=(c/(2f))*N ,
where c is the speed of light, f – the frequency, the integer N is the order of the peak.

We added this part to the lines 127-129.

Reviewer 2 Report

In this manuscript, the authors report on a detailed theoretical study of a THz detector device based on thermoelectric materials. The results are sound and the manuscript is well written and organized. I would therefore recommend publication of this manuscript in Applied Sciences. Prior to publication, some improvements may be nevertheless performed as follows:

1). The number of references regarding the thermoelectric properties of Sb and Bi1-xSbx is rather low. Considering the numerous studies performed on the thermoelectric properties of these conventional TE materials, additional references should be added in the reference list.

2). The authors have chosen these two materials to perform their simulations. However, their choice in not enough well justified in the manuscript. Is it more a question of the presence of a narrow band gap (or a semi metallic state) in the materials or a question of responsivity of the device depending on the thermopower values? In this regard, may charge-density-wave compounds be used for such devices? If only the thermopower values matter, may other TE materials with higher thermopower be considered? Some comments about these points in the introduction or conclusion would be interesting. 

3). Figures 3 to 5 are not commented in the manuscript. The reader would benefit from a clear and concise discussion of these results.

Author Response

1). The number of references regarding the thermoelectric properties of Sb and Bi1-xSbx is rather low. Considering the numerous studies performed on the thermoelectric properties of these conventional TE materials, additional references should be added in the reference list.

    Re: We added more links and data about thermoelectric properties of thin films based on Bi-Sb(x) solid solutions. Now, there are links in the text to the works of Volklein [19], Suslov [20], Cho [21] and Linseis [22].

2). The authors have chosen these two materials to perform their simulations. However, their choice in not enough well justified in the manuscript. Is it more a question of the presence of a narrow band gap (or a semi metallic state) in the materials or a question of responsivity of the device depending on the thermopower values? In this regard, may charge-density-wave compounds be used for such devices? If only the thermopower values matter, may other TE materials with higher thermopower be considered? Some comments about these points in the introduction or conclusion would be interesting. 

    Re: Both Seebeck and and gap are of great importance. We added additional referencses to show the advantage of Bi-Sb at room temperature from the point of view of Seebeck (lines 36-43). We added several references to show that Bi-Sb has narrow band gap (lines 43-47, references [24-26]) that point to the fact that this compound must be sensitive to THz radiation.

         Other charge-density-wave thermoelectric compounds may be sensitive to THz too. Up to the moment, it's not studied enough. In conclusion (lines 154-156) we resume that other thermoelectrics with close properties  potentially can be used as the basis of THz sensors.

3). Figures 3 to 5 are not commented in the manuscript. The reader would benefit from a clear and concise discussion of these results

    Re: We added the analysis of properties change in THz frequency range (Section 2.3 Materials, lines  97-107), as well as more details of data processing (equations 15-17), comparison of the results with Drude model (equations 18-19).